# UV Light-Emitting-Diode Traps for Collecting Nocturnal Biting Mosquitoes in Urban Bangkok

**DOI:** 10.3390/insects13060526

**Published:** 2022-06-07

**Authors:** Suntorn Pimnon, Ratchadawan Ngoen-Klan, Anchana Sumarnrote, Theeraphap Chareonviriyaphap

**Affiliations:** 1Department of Entomology, Faculty of Agriculture, Kasetsart University, Bangkok 10900, Thailand; suntorn.pi@ku.th (S.P.); grrwn@ku.ac.th (R.N.-K.); 2Department of Entomology, Faculty of Agriculture at Kamphaeng Saen, Kamphaeng Saen Campus, Kasetsart University, Nakhon Pathom 73140, Thailand; fagraas@ku.ac.th

**Keywords:** light traps, fluorescent, *Culex*, *Aedes*, Thailand

## Abstract

**Simple Summary:**

This study was conducted to evaluate the efficacy of six ultraviolet light-emitting diodes (UV-LED) traps and a fluorescent light trap for sampling urban nocturnal mosquitoes. Results demonstrated that the fluorescent light trap outperformed all the UV-LED traps throughout the 72 sampling nights and between wet and dry seasons. Among the UV-LED traps, the LED375 trapped the highest number of mosquitoes. Additional field trials are needed to validate these findings in different ecological settings.

**Abstract:**

Well-designed surveillance systems are required to facilitate a control program for vector-borne diseases. Light traps have long been used to sample large numbers of insect species and are regarded as one of the standard choices for baseline insect surveys. The objective of this study was to evaluate the efficacy of six ultraviolet light-emitting diodes and one fluorescent light for trapping urban nocturnal mosquito species within the Kasetsart University (KU), Bangkok. Ultraviolet light-emitting diodes (UV-LEDs), (LED365, LED375, LED385, LED395, and LED405) and a fluorescent light were randomly assigned to six different locations around the campus in a Latin square design. The traps were operated continuously from 18:00 h to 06:00 h throughout the night. The traps were rotated between six locations for 72 collection-nights during the dry and wet seasons. In total, 6929 adult mosquitoes were caught, with the most predominant genus being *Culex*, followed by *Aedes*, *Anopheles*, *Armigeres* and *Mansonia*. Among the *Culex* species, *Culex quinquefasciatus* (*n* = 5121: 73.9%) was the most abundant followed by *Culex gelidus* (*n* = 1134: 16.4%) and *Culex vishnui* (*n* = 21: 0.3%). Small numbers of *Aedes*, *Armigeres,* and *Anopheles* mosquitoes were trapped [*Aedes albopictus* (*n* = 219: 3.2%), *Aedes pocilius* (*n* = 137: 2.0%), *Armigeres subalbatus* (*n* = 97: 1.4%), *Anopheles vagus* (*n* = 70: 1.0%), *Aedes aegypti* (*n* = 23: 0.3%)]. There were 2582 specimens (37.2%) captured in fluorescent light traps, whereas 942 (13.6%), 934 (13.5%), 854 (12.3%), 820 (11.8%), and 797 (11.5%) were captured in the LED375, LED405, LED395, LED365, and LED385 traps, respectively. None of the UV-LED light traps were as efficacious for sampling nocturnal mosquito species as the fluorescent light trap. Among the five UV-LED light sources, LED375 trapped the greatest number of mosquitoes. Additional field trials are needed to validate these findings in different settings in order to substantially assess the potential of the LEDs to trap outdoor nocturnal mosquitoes.

## 1. Introduction

Vector control is a key component of disease control and elimination. Several tools are under development to determine distribution, abundance, and infection rate of the mosquito, among which various traps are of interest for mosquito sampling and surveillance [1]. Light traps have been used for a long time as a common basic surveying equipment device for insects, and there have been many variations in the design of light traps over the decades. Although its use in urban environments is facilitated by the availability of nearby power sources, field applications are often limited by the requirement for electricity to power traditional lights. Traditional fluorescent bulbs usually do not run for more than 12 h from a conventional 12-volt power source, such as a car battery [2]. Light traps have been used to trap insects for over 100 years [3,4] with many different types of designs, with some being very complicated, involving both the lights and fans, while others have remained uncomplicated [5].

Most current traps use incandescent bulbs or actinic fluorescent traps as the light source because the spectrum of light emitted by these bulbs is effective in attracting insects [6,7]. However, maintaining the necessary power to illuminate these light sources is always an issue [8]. Typically, small bulbs of around 6–9 watts will require either a fixed mains-connected power source or a large portable power supply to provide illumination for the entire night. A typical power source is a 12-volt battery, which will provide approximately 6–8 h, depending on the type of battery. Due to the flight period of various insects varying from dusk to dawn, standard light sources might not be able to attract some of the existing insect population.

Nocturnal insects are attracted by artificial light sources. Recently, light sources that produce large amounts of ultraviolet light-emitting diode (UV-LED) light trap radiation have revolutionized light trap sampling. Light traps are generally expensive, but some are very effective for the collection of insects [9,10]. Various light sources have been used for sampling purposes, such as mercury vapor lamps, gas lamps, and fluorescent UV light traps [11]. The UV light trap (LT trap) was designed using black plastic material equipped with an electric fan and an artificial light source supplied by 220 V electricity [12,13]. However, there are three types of UV radiation: UVA, UVB, and UVC with wavelengths in the ranges of 315–400 nm, 280–315 nm, and 100–280 nm, respectively. These three types of UV radiation are grouped according to wavelength, which varies according to insect biological activity [14,15]. UVA (315–400 nm) consists of long-wave ultraviolet or black light and is not absorbed by the ozone layer, hence it is a safe wavelength for users [16]. Numerous studies have used and evaluated LED for trapping mosquitoes [17]. However, no study has been published that compared the different UVA wavelengths in trapping nocturnal biting mosquitoes in Thailand. To address these shortcomings, different spectral ranges of UVA (365, 375, 385, 395, and 405 nm) were evaluated by comparison with a fluorescent light trap source for collecting mosquitoes in Bangkok, Thailand.

## 2. Materials and Methods

### 2.1. Study Sites

This study was conducted at the Kasetsart University (KU) (13°50′32.96″ N, 100°34′2.98″ E), Bangkok, Thailand. The campus covers 135.7 hectares and is categorized as an urban area. The campus consists of either buildings or natural sites that provide an ideal breeding habitat for mosquitoes. In total, six square grids (approximately 1.3 km^2^/grid) were overlaid on a map of the campus and used as six study locations: Chobprachoom Building, Faculty of Fisheries (Loc. 1), Faculty of Agriculture (Loc. 2), Ngamwongwan 1 Parking Garage (Loc. 3), KU Dormitory for male students (Loc. 4), the Office of Agricultural Museum and Culture (Loc. 5) and the Entomology Research Building (Loc. 6), as shown in (Figure 1). Each grid was subdivided into six smaller squares in which trap locations were randomly selected.

### 2.2. Trapping Method

Black Hole™ Mosquito trap units (Pan Science Co., Ltd., Bangkok, Thailand) were used as the reference trapping method. Briefly, the trap was made of durable black plastic material equipped with an electrical fan and used a fluorescent lamp as the ultraviolet light (UV) source in the range of 100–400 nm. As an alternative source of UV light to this fluorescent lamp, LED UV light traps purchased from a department store were investigated. The LED UV lights with titanium dioxide (TiO_2_) produced carbon dioxide (CO_2_) and had 5 spectral lines of UV light-emitting diodes (UV-LEDs) consisting of 365 nm (LED365), 375 nm (LED375), 385 nm (LED385), 395 nm (LED395) and 405 nm (LED405). These were examined for their effectiveness in sampling adult mosquitoes compared to the Black Hole™ Mosquito trap. This trap consisted of on O-ring hanging with standalone columns or beams in a box that operated without any spark or noise. Each trap was hung, approximately 1.5 m above the ground.

### 2.3. Experimental Design

A Latin square design was applied in this study to minimize the residual error in the experiment by eliminating variance due to any known and controllable disturbance variables [18]. The light traps for catching adult mosquitoes were rotated through the 6 sites based on a 6 × 6 Latin square design, where one replication comprised 6 consecutive nights of trapping. The experiments were performed for 6 replications (36 nights) each in the dry (February to April 2020) and wet (July to September 2020) seasons.

### 2.4. Mosquito Collection

A Nocturnal mosquito species were captured from the six light traps which were operated continuously from 18:00 to 06:00 h. All captured mosquitoes were removed from light traps every three hours at 21:00 h, 24:00 h, 03:00 h, and 06:00 h Morphological identification of mosquito species was performed following the standard illustrated keys to adult mosquitoes [19,20,21,22,23] the next morning. To confirm the mosquito species, larvae or pupae were sampled once nearby a trap position during a season and reared to adult mosquitoes in the insectary at the Department of Entomology, Faculty of Agriculture, Kasetsart University. The meteorological data (relative humidity, temperature, and rainfall) were recorded using climatological data for the period between 2020 and 2021 from Don Muang Airport Station, Bangkok.

### 2.5. Data Analysis

The total number of mosquitoes captured per light trap per night was transformed using the logarithm function (Log10(x + 1)) to normalize the distribution of the one-way analysis of variance (ANOVA), and the Tukey’s test for post hoc analysis was performed to evaluate the efficacy of the five UV-LEDs (LED365, LED375, LED385, LED395, and LED405) compared to the UV fluorescence. The percentage of mosquito species caught per trap per night was calculated by dividing the number of mosquito species from an individual light trap by the total number of mosquito species collected in the night and multiplying by 100. Mean percentages of mosquito species captured per trap per night were analyzed using the one-way ANOVA and Tukey’s test for post-hoc analysis.

A generalized linear model (GLM), consisting of a negative binomial model and a log link function, was used to analyze the main parameter of the light source and the co-parameter of the night collection that influenced the numbers of mosquitoes collected per trap per night in each season. Parameter coefficients were evaluated using the Wald chi-square test. Incident rate ratios (IRR) of the different light sources were calculated relative to the reference light trap of the UV fluorescence. Values of IRR greater or less than 1 indicated higher or lower trapping performance, respectively, relative to the reference to determine whether each of the alternative light sources (UV-LEDs) was correlated with the referenced light source (UV fluorescent). The Pearson’s correlation coefficient was used to investigate the relationship among the log-transformed catches for each mosquito species. All data were analyzed using the SPSS Statistics for Windows software, version 26.0 (IBM Corp, Armonk, NY, USA) with a significance level of 0.05.

## 3. Results

Five LED-UV traps with different spectral ranges (365–405 nm) were compared with one fluorescent light source for efficacy in trapping urban mosquito species. Traps were set for 72 nights collection in both the dry and wet seasons. In total, 6929 adult mosquitoes were recorded in the trap types: fluorescent UV (35.64%, 34.72%), LED405 (14.76%, 12.35%), LED375 (13.04%, 14.78%), LED395 (12.96%, 12.38%), LED385 (12.86%, 11.74%), and LED365 (11.15%, 14.52%), as shown in (Figure 2).

Species within five genera were morphologically identified as *Ae. aegypti* (*n* = 23: 0.3%), *Ae. albopictus* (*n* = 219: 3.2%), *Ae. pocilius* (*n* = 137: 2.0%), *An. vagus* (*n* = 70: 1.0%), *Ar. subalbatus* (*n* = 97: 1.4%), *Cx. gelidus* (*n* = 1134: 16.4%), *Cx. quinquefaciatus* (*n* = 5121: 73.9%), *Cx. vishnui* (*n* = 21: 0.3%), *Mansonia uniformis* (*n* = 28: 0.4%), and other species (*n* = 79: 1.1%). The highest number of mosquitoes was collected from fluorescent-UV light traps compared to the other five LED-UV light sources. In terms of abundance, the fluorescent light traps captured the greatest number of mosquitoes (*n* = 2582; 37.2%), followed by LED375 (*n* = 942; 13.6%), LED405 (*n* = 934; 13.5%), LED395 (*n* = 854; 12.3%), LED365 (*n* = 820; 11.8%), and LED385 (*n* = 797; 11.5%), respectively (Table 1). There were no significant differences in the mean numbers of collected mosquitoes from each light source between the two seasons during the collection period (Table 2).

The efficacy of the light traps at catching the two *Culex* species is provided in Table 3. Overall, the greatest number of mosquitoes were collected from the UV fluorescence light traps (38.4% in dry and 37.7% in wet), regardless of the season. Specifically, a higher number of *Cx. quinquefasciatus* was caught during the dry season compared to the wet season, whereas there was a higher number of *Cx. gelidus* caught in the wet season. There was no significant difference in the mean number of mosquitoes caught from each trap source between the two seasons for both *Culex* species, except those collected from UV fluorescence light trap, where there was a significant difference between seasons.

A negative binomial regression GLM was performed to determine whether two key factors (light source, season) influenced the efficacy of light traps in capturing nocturnal mosquito species. From the goodness-of-fit test, the deviance (1.033) and Pearson chi-square (443.689) demonstrated that the negative binomial regression was perfectly suitable (Omnibus test; *p* = 0.000). Based on results from Table 4, only the light source parameter was a significant predictor that influenced the number of mosquitoes captured per trap, while seasons and nights were not significant variables in the prediction model (*p* = 0.45). For the fluorescent UV set as the reference (IRR = 1), the IRR values for LED375, LED405, LED395, LED365, and LED385 were 0.364, 0.362, 0.331, 0.327, and 0.307, respectively. These results indicated that the fluorescent UV was the most efficient light source to capture mosquitoes compared to the other LED light sources (Table 4).

The experiment was conducted during two seasons: dry (February–April) and wet (July–September). The meteorological data indicated that there was higher rainfall between July and October than for the other months of the year. The highest rainfall was in July (287.6 mm), whereas no rainfall (0 mm) was recorded in February and December. The mean relative humidity and temperature were also recorded (Figure 3).

Larval collections made in the vicinity of trap placements identified six species belonging to four genera (*Culex*, *Aedes*, *Anopheles*, *Toxorhynchites*), the highest number of *Cx. quinquefasciatus* (*n* = 815: 96.5%), *Cx.*
*gelidus* (*n* = 7: 0.8%), *Lutzia (Metalutzia) fuscana* (*n* = 2: 0.2%), *Ae.*
*albopictus* (*n* = 12: 1.4%), *Ae.*
*aegypti* (*n* = 1: 0.1%) and *Anopheles barbrirostris* (*n* = 4: 0.5%). 

Two specimens of *Anopheles* (*n* = 2: 0.2%) and two from *Toxorhynchites* (*n* = 2: 0.2%) could not be identified due to specimen damage (Figure 4).

## 4. Discussion

Mosquito traps are generally used in surveillance to monitor the distribution and abundance of mosquito populations [24]. Recently, several traps have been developed not only for surveillance or monitoring but also for vector control. A light trap is one type of trap that is popular for trapping mosquitoes [25]. The several types of light sources of LED lights have demonstrated efficacy in attracting various insects and pests [26,27,28,29,30]. Although some studies using light traps have been reported in Thailand [31,32], no known study has been published on the use of different wavelengths of LED lights for collecting nocturnally active urban mosquitoes in Thailand. Therefore, we compared the efficacy of five different wavelengths of UV-LED light sources to catch nocturnal mosquitoes in an urban environment of Bangkok, Thailand. 

In this study, fluorescent lights outperformed all five different wavelengths of LED lights in catching urban mosquito species. Overall, nine nocturnal mosquito species were collected, with the most predominant species being *Cx. quinquefasciatus* (73.9%) across all collections, regardless of the light configuration, trap location, or collection date. The potential of *Cx. quinquefasciatus* as a vector in urban areas is further indicated by its typical breeding underground in sewers and drains [33]. This species is a very common, cosmopolitan urban nighttime biting mosquito and is generally active during the entire evening, depending on the availability of vertebrate hosts [34,35,36,37]. *Culex quinquefasciatus* is a primary vector of lymphatic filariasis disease, which has caused serious negative social, economic, and health impacts in tropical and subtropical countries [38]. A report in Thailand revealed that this mosquito species is able to transmit several other medically important pathogens, such as Japanese encephalitis, West Nile virus, Zika virus, and Tembusu virus [39].

Kasetsart University is located in a heavily urbanized area of the Bangkok metropolis. The university grounds provide many suitable larval habitats covering large bodies of open stagnant/slow moving water, numerous rainwater drainage lines, and various artificial containers infused with varying degrees of organic matter and higher levels of pollution that provide suitable breeding areas in different environments. *Culex gelidus* was the second-most common species (12.6%) captured in all traps. This species is of particular interest, as it is a natural vector of Japanese encephalitis virus (JEV) between host birds and humans [40]. The campus is also home for both migratory and resident wild bird species (Family Ardeidae: egrets, herons and bitterns), that are potential reservoirs of JEV. In addition, the campus is normally congested with human activity during the day and at nighttime; thus, virus transmission to humans is a possibility.

The results demonstrated that the traps may not be species-specific, perhaps suggesting that other factors, such as mosquito density, could influence the trapping efficacy for a particular species. However, abundance estimates of mosquito species and their relative composition provided by different trapping devices could provide beneficial information for guiding surveillance methods and control efforts.

Differences between the two seasons in indices, such as ambient temperature, relative humidity, and rainfall, could be the factors that greatly influence adult mosquito activity and behavior [41]. Furthermore, the moon phase (waxing and waning) may influence the numbers of mosquitoes trapped due to the effect of the moonlight intensity and duration of illumination. The effect of the moonlight and lunar periodicity on light trap catches of mosquito species has been described by several authors [42,43]. Moonlight appears to reduce the number of mosquito collected from light traps under certain circumstances; for example, possibly by providing competitive illumination between the brightness (intensity) of the trap light source against background illumination resulting from the moon and thereby decreasing the contrast (attractiveness) by reducing the area in which mosquitoes are drawn to the trap [44]. Although the moon phase was recorded for each collection night throughout the study, several other factors influencing catch size and the study design itself precluded quantifying the possible effects of moonlight on catch size [45,46,47].

The first mosquito survey trap developed in the 1930s (the New Jersey light trap) remains among the most productive and efficient traps available for mosquito surveillance [5,48,49]. Several types of mosquito trapping devices have been developed and utilized over the years for mosquito surveillance [50,51,52]. Centers for Disease Control and Prevention (CDC) miniature light traps of different designs have been the standard used in at least one other study to conduct mosquito surveillance in Thailand [17]. Various devices, such as CDC traps baited with CO_2_ (or other semiochemicals) and Biogents (BG) lures with a combination of olfactory attractants, are available for adult mosquito sampling [51,53]. Recently, new traps using different light sources, modified designs, and attractors have been developed and evaluated against CDC light traps. However, there are no known studies published that have evaluated the attractive efficacy of LED illumination compared to fluorescent UV light. Additionally, the collection of outdoor active mosquitoes is limited in terms of the effectiveness of trapping devices. The present study in KU found that a fluorescent UV light trap achieved the greatest yield of attracted mosquitoes. Previous studies have demonstrated that among the five LEDs in the current study, the fluorescent UV light wavelength was an effective attractant for capturing mosquitoes [54,55,56,57]. Operationally, the Black Hole™ Mosquito trap was an acceptable device for mosquito collection. However, there are limits to its application for placement and position due to the need to for direct current electricity (from the main electricity distribution system) compared to other traps powered by batteries. Consequently, LED traps are increasingly used in mosquito traps since they have several advantages, including energy efficiency, durability, long lifetime, and good temporal stability.

This study was conducted to evaluate the attractiveness of different specific UVA wavelengths in trapping nocturnal mosquito species. As previously reported, mosquito species were attracted by different specific wavelengths of LED lights [27]. Wild *Culex. pipiens* has been reported as a species closely related to *Cx. quinquefasciatus* [58], which responded differently depending on the wavelengths of the light source. The most effective wavelengths to attract this species were between 333 nm and 405 nm in near UV wavelengths [59]. UV light can be categorized into three groups based on wavelength, UVA (315–400 nm), UVB (280–315 nm), and UVC (100–280 nm) [60]. Other publications have reported that UVA displayed high relative efficacy in attracting and trapping nocturnal mosquitoes [61,62]. Observations based on an electroretinogram have shown the highest responses of *Cx. pipiens* to 335 nm, corresponding to the UVA range (315–400 nm) [63].

One of the primary goals for understanding mosquito biology and ecology is measuring mosquito populations and species dynamics to facilitate the design and implementation of appropriate prevention and control strategies. An in-depth evaluation and analysis of a mechanical light trapping system for attracting nocturnally active mosquitoes can provide important information for conducting mosquito surveillance. The current study was the first attempt to assess various light sources as mosquito attractants in a densely populated urban area of Bangkok, while also obtaining information on mosquito species present at the Kasetsart University. Studies are continuing at the same location to evaluate the attractive responses to different UV wavelengths and trapping systems.

## 5. Conclusions

None of the tested UV-LED light traps were as efficacious for sampling nocturnal mosquito species as the fluorescent light trap. There were no significant differences in the numbers of collected mosquitoes from each LED light source. Among the five UV-LEDs, LED375 trapped the greatest number of mosquitoes. The comparative trapping efficacy of all light sources did not vary with season. However, owing to the efficacy and advantages of the LED light traps, they could have potential for mosquito surveillance as well as vector control. Additional field trials are needed to validate these findings in different settings and to assess the effectiveness of different wavelengths for LED light sources in trapping mosquito species.

## Figures and Tables

**Figure 1 insects-13-00526-f001:**
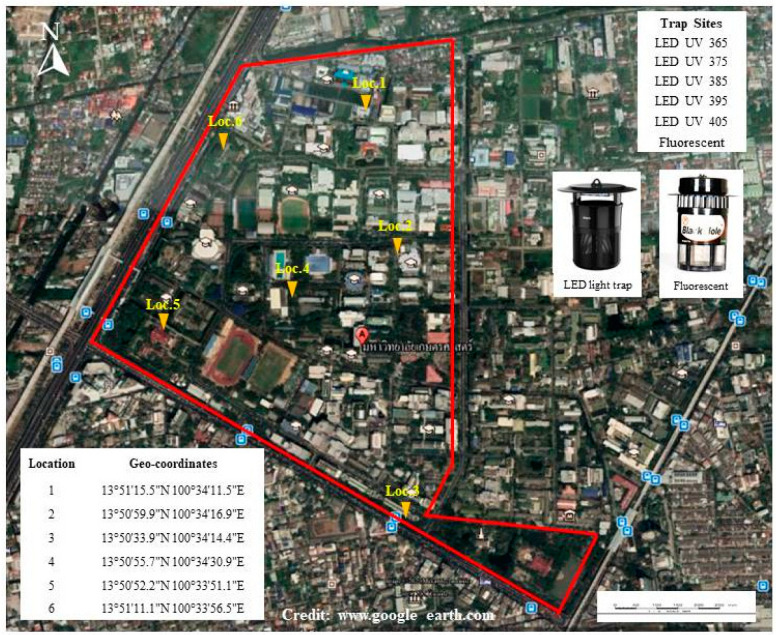
Locations for setting light traps (18:00 to 06:00 h) to capture adult mosquitoes at Kasetsart University, Bangkok, Thailand.

**Figure 2 insects-13-00526-f002:**
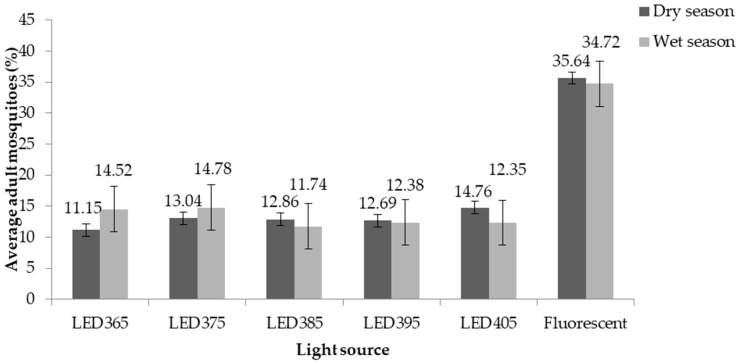
Percentage of mosquitoes trapped using different light source traps between dry and wet seasons.

**Figure 3 insects-13-00526-f003:**
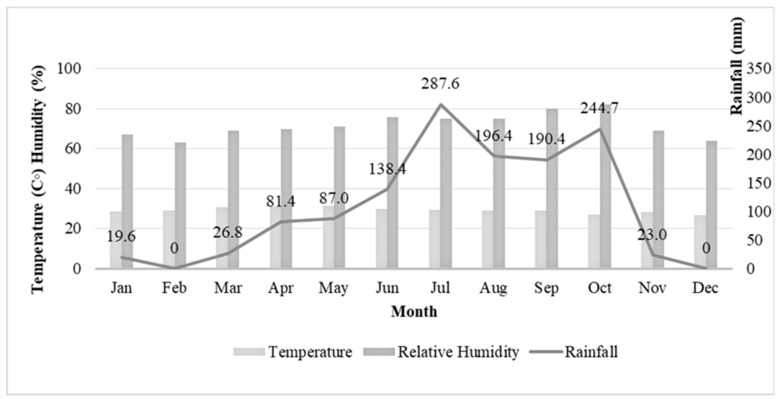
Monthly environmental parameters (mean temperature, relative humidity, and total rainfall) for study area.

**Figure 4 insects-13-00526-f004:**
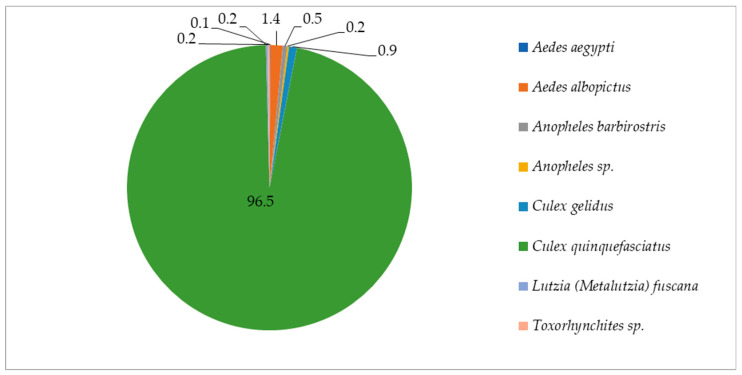
Proportion of mosquito species obtained from larval collections of nearby trap locations.

**Table 1 insects-13-00526-t001:** Total number and percentage of adult mosquito species captured in traps with 6 different light sources over 6 replications (6 nights/replication) in dry and wet seasons at Kasetsart University, Bangkok.

Trap Light Source	Collection Nights	Total Number of Mosquito Species (%)										
		*Ae. aegypti*	*Ae. albopictus*	*Ae. pocilius*	*An. vagus*	*Ar. subalbatus*	*Cx. quinquefasciatus*	*Cx. gelidus*	*Cx. vishnui*	*Ma. uniformis*	Others *	Total
LED365	72	3	29	36	16	5	519	183	6	5	18	820 (11.8)
LED375	72	3	35	11	7	6	743	127	0	3	7	942 (13.6)
LED385	72	8	22	17	12	36	576	102	6	5	13	797 (11.5)
LED395	72	2	25	18	14	13	639	132	0	1	10	854 (12.3)
LED405	72	3	29	21	8	5	754	100	3	4	7	934 (13.5)
Fluorescent	72	4	79	34	13	32	1890	490	6	10	24	2582 (37.3)
**Total**		23(0.3)	219(3.2)	137(2.0)	70(1.0)	97(1.4)	5121(73.9)	1134(16.4)	21(0.3)	28(0.4)	79(1.1)	6929(100)

* These specimens could not be identified to the species level due to damage and insufficient numbers to make comparisons.

**Table 2 insects-13-00526-t002:** Mean number of mosquitoes obtained from 6 light sources over 6 replications (6 nights/replication) at Kasetsart University, Bangkok between dry and wet seasons.

Trap Light Source	Collection Nights Dry/Wet	Total (%) Dry/Wet	Mean ± SD *		95% Confidence Interval			
			Dry **	Wet **	Dry		Wet	
					Lower	Upper	Lower	Upper
LED365	36/36	365 (10.1)/ 832 (22.5)	0.85 ± 0.07 ^a^	0.95 ± 0.07 ^a^	0.70	1.01	0.81	1.09
LED375	36/36	480 (13.2)/ 462 (12.5)	0.09 ± 0.07 ^a^	0.97 ± 0.06 ^a^	0.74	1.06	0.83	1.10
LED385	36/36	449 (12.3)/ 348 (9.4)	0.91 ± 0.07 ^a^	0.88 ± 0.06 ^a^	0.76	1.06	0.76	1.00
LED395	36/36	441 (12.1)/ 440 (11.9)	0.85 ± 0.08 ^a^	0.92 ± 0.06 ^a^	0.68	1.02	0.78	1.06
LED405	36/36	562 (15.5)/ 372 (10.1)	0.97 ± 0.08 ^a^	0.96 ± 0.04 ^a^	0.80	1.13	0.87	1.06
Fluorescent	36/36	1339 (36.8)/ 1243 (33.6)	1.40 ± 0.06 ^a^	1.41 ± 0.05 ^a^	1.27	1.53	1.29	1.53

* Mean number of log-transformed data ± SD of all adult mosquitoes captured by each light source trap carried out during 36 sampling nights in dry and wet seasons. ** Values with the same lowercase superscripts in a column for each season are not significantly different using one-way ANOVA with a multiple Tukey’s test comparison at the 0.05 level.

**Table 3 insects-13-00526-t003:** Mean numbers of *Cx. quinquefasciatus* and *Cx. gelidus* collected during 36 trapping nights in dry and wet seasons using 6 different light source traps.

Trap Light Source	Night Dry/Wet	Total (%) Dry/Wet	Mean ± SD **			
			*Cx. quinquefasciatus*		*Cx. gelidus*	
			Dry *	Wet *	Dry *	Wet *
LED365	36/36	296 (9.0)/ 406 (13.6)	7.89 ± 1.44 ^a^	6.53 ± 1.15 ^a^	0.33 ± 0.15 ^a^	4.74 ± 1.05 ^a^
LED375	36/36	453 (13.8)/ 417 (14.0)	12.31 ± 2.82 ^a^	8.33 ± 1.55 ^a^	0.28 ± 0.13 ^a^	3.25 ± 0.65 ^a^
LED385	36/36	367 (11.2)/ 311 (10.4)	9.69 ± 1.98 ^a^	6.31 ± 1.30 ^a^	0.50 ± 0.22 ^a^	2.33 ± 0.40 ^a^
LED395	36/36	381 (11.6)/ 390 (13.1)	10.44 ± 2.07 ^a^	7.31 ± 1.54 ^a^	0.14 ± 0.58 ^a^	3.53 ± 1.05 ^a^
LED405	36/36	522 (15.9)/ 332 (11.1)	14.17 ± 2.93 ^a^	6.78 ± 0.74 ^a^	0.33 ± 0.13 ^a^	2.44 ± 0.57 ^a^
Fluorescent	36/36	1258 (38.4)/ 1122 (37.7)	33.92 ± 6.11 ^b^	18.58 ± 1.77 ^a^	1.03 ± 0.33 ^a^	12.58 ± 3.42 ^b^

** Mean numbers (± SD) of all adult mosquitoes captured in each light source trap carried out during 36 sampling nights in dry and wet seasons. * Values in each column with different lowercase superscripts are significantly different using one-way ANOVA with a multiple Tukey’s test comparison at the 0.05 level.

**Table 4 insects-13-00526-t004:** Incidence rate ratios of factors influencing efficacy of light traps to capture nocturnally active mosquitoes.

Parameter Estimate										
Parameter	B	Std. Error	95% Wald Confidence Interval		Hypothesis Test			IRR	95% Wald Confidence Interval for Exp(B)	
			Lower	Upper	Wald Chi-Square	df	Sig.		Lower	Upper
(Intercept)	3.611	0.1589	3.300	3.922	516.572	1	0.000	37.003	27.102	50.521
LED365	−1.119	0.1720	−1.456	−0.782	42.309	1	0.000	0.327	0.233	0.458
LED375	−1.010	0.1710	−1.345	−0.675	34.910	1	0.000	0.364	0.260	0.509
LED385	−1.181	0.1716	−1.518	−0.845	47.387	1	0.000	0.307	0.219	0.430
LED395	−1.105	0.1714	−1.441	−0.769	41.598	1	0.000	0.331	0.237	0.463
LED405	−1.017	0.1716	−1.353	−0.681	35.138	1	0.000	0.362	0.258	0.506
Fluorescent UV	0 ^a^							1		
Dry season	0.076	0.1003	−0.121	0.272	0.572	1	0.449	1.079	0.886	1.313
Wet season	0 ^a^							1		
Night	−0.021	0.0283	−0.077	0.034	0.570	1	0.450	0.979	0.926	1.035
(Scale)	1 ^b^									
(Negative binomial)	1 ^b^									

^a^ Set to zero because this parameter is redundant. ^b^ Fixed at displayed value.

## Data Availability

All relevant data are included in the article.

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
