# Peer review of "UV Light-Emitting-Diode Traps for Collecting Nocturnal Biting Mosquitoes in Urban Bangkok"

_insects, 2022, doi:10.3390/insects13060526_

Round 1

Reviewer 1 Report

The manuscript entitled “UV light-emitting diode traps for collecting nocturnal biting mosquitoes in urban Bangkok” aimed to evaluate LED traps for trapping mosquitoes, but I unfortunately detected some unacceptable and unethical practices on it.

This paper is full of false citations. Citations accredited to an author or a paper but when I search for that citation in the original paper, I cannot find such citations neither as TEXT not as IDEA. It must be prevented.

EXAMPLE: Please verify the citation # 18, nothing about what authors say (Latin square delineation) - Microsoft Word - DistributionFitting_2.doc (arxiv.org).

Authors have written a sentence, but when I search for this sentence or its idea in the original paper, I cannot find both. Authors must differentiate true citations from false ones.

False citations will be abandoned.

Some examples of super false citations:

Citation # 5 - deals with Saccharomyces cerevisiae plasmid (; #11 – on genome sequencing, published on Nature; # 23 – published on Nanotechnology; #31 – published on nanoscale; etc..

These malpractices should be curbed.

Author Response

Response to Reviewer 1 Comments

Point 1: The manuscript entitled “UV light-emitting diode traps for collecting nocturnal biting mosquitoes in urban Bangkok” aimed to evaluate LED traps for trapping mosquitoes, but I unfortunately detected some unacceptable and unethical practices on it.

Point 2: This paper is full of false citations. Citations accredited to an author or a paper but when I search for that citation in the original paper, I cannot find such citations neither as TEXT not as IDEA. It must be prevented.

Point 3: EXAMPLE: Please verify the citation # 18, nothing about what authors say (Latin square delineation) - Microsoft Word - DistributionFitting_2.doc (arxiv.org).

Point 4: Authors have written a sentence, but when I search for this sentence or its idea in the original paper, I cannot find both. Authors must differentiate true citations from false ones.

Point 6: Some examples of super false citations:

Point 7: Citation # 5 - deals with Saccharomyces cerevisiae plasmid (; #11 – on genome sequencing, published on Nature; # 23 – published on Nanotechnology; #31 – published on nanoscale; etc.

Response: Thank you very much for the comments and suggestions. All false citations have been worked upon in text and references list. Non-corresponding texts have been removed and replaced suitably.

Citation #5: Muirhead-Thompson, R. Trap responses of flying insects: the influence of trap design on capture efficiency. 2012.

Citation #11: Tamuri, A.; Muhamad, A.; Akmal, S.; Lani, M.; Kundwal, M.; Daud, Y. Ultravoilet (UV) Light Spectrum of flourescent lamps; 2014.

Citation #23: Bentley, M.T.; Kaufman, P.E.; Kline, D.L.; Hogsette, J.A. Response of adult mosquitoes to light-emitting diodes placed in resting boxes and in the field. Journal of the American Mosquito Control Association 2009, 25, 285-291.

Citation #31: Uttah, E.C.; Wokem, G.N.; Okonofua, C. The abundance and biting patterns of Culex quinquefasciatus Say (Culicidae) in the coastal region of Nigeria. International Scholarly Research Notices 2013, 2013.

Response: Thank you. Revised accordingly.

Reviewer 2 Report

This study aimed to evaluate the effectiveness of fluorescent light traps for collecting mosquitoes in Bangkok, Thailand. The manuscript is well written and the methodology is appropriate to test their hypothesis. I have only minor suggestions to help improve the manuscript.

Please mention in the introduction and include as a limitation in the discussion that fluorescent traps have different levels of attractiveness to different mosquito species. Also, collecting mosquitoes from 18:00 to 06:00 is not indicated to collect important diurnal species such as Aedes aegypti and Aedes albopictus. Even though the results are solid and the analyses are appropriate, this is an intrinsic limitation of light-based traps. 

Please correct minor typos in the text. Also italicize all scientific names. 

Figures should also be improved, if possible. Consider improving the resolution and presentation of the figures. For example, in figure 1 the cartesian arrow could have its white background removed; the number above the bars in figure 2 are overlapping. 

Table 1. Please improve the layout of the table. In its current form, it is very difficult to understand what's been shown. 

Author Response

Response to Reviewer 2 Comments

Point 1: This study aimed to evaluate the effectiveness of fluorescent light traps for collecting mosquitoes in Bangkok, Thailand. The manuscript is well written and the methodology is appropriate to test their hypothesis. I have only minor suggestions to help improve the manuscript.

Response: Thank you.

Point 2: Please mention in the introduction and include as a limitation in the discussion that fluorescent traps have different levels of attractiveness to different mosquito species. Also, collecting mosquitoes from 18:00 to 06:00 is not indicated to collect important diurnal species such as Aedes aegypti and Aedes albopictus. Even though the results are solid and the analyses are appropriate, this is an intrinsic limitation of light-based traps. 

Response: Text insertion with reference has been done.

Point 3: Please correct minor typos in the text. Also italicize all scientific names. 

Response: Corrected.

Point 4: Figures should also be improved, if possible. Consider improving the resolution and presentation of the figures. For example, in figure 1 the cartesian arrow could have its white background removed; the number above the bars in figure 2 are overlapping. 

Response: Figure 1 the white background of the cartesian pointer has removed background. Figure 2 the numbers have been moved up to make the numbers clearer.

Point 5: Table 1. Please improve the layout of the table. In its current form, it is very difficult to understand what's been shown. 

Response:  The layout for table 1 has been edited.

Response: Thank you. Revised accordingly.

Reviewer 3 Report

This is a very well written manuscript with a comprehensive experimental design and relevant analysis. The article does need to be combed through for standardizing acronym use throughout.

Line 43-45, Line 49-51: This reads redundantly for me. I would revisit these two sentences and combine/revise.

Line 59: It looks like this should be “, depending on the type of battery.”

Line 63: Spell out UV on first use

Line 66: Are you using LT trap throughout the document? Either remove this or standardize throughout.

Line 73: Spell out LED on first use

Line 84: Signify urban mosquitoes

Figure 1: Is it supposed to say “trap sites” above the light types?

Line 101: Lowercase T in trap

Line 129: Lowercase O in one-way

Figure 2: This needs some reformatting. I am not sure if including the values are necessary in this graph (they are visualized by the y-axis and written in text) but if the authors with to include them then the values should not overlap each other or the error bars.

Table 1: This table is cluttered and difficult to visualize. Add some space between mosquito species. “Trap light source” and “Collection nights” labels are too cramped.

Table 2 & 3: This table needs revisiting. It is unclear what “*”, “**”, “***”, “a” and “b” means.

Line 175: Italics Cx. quinquefasciatus

Line 176: Italics Cx. gelidus

Line 217-218: I suggest this sentence be removed; it appears to suggest that light traps can be used to reduce vector densities. If the author is suggesting that this is the case, then please provide a citation that this is possible.

Line 223: If you are beginning a sentence, spell out the entire genus.

Line 270: Spell out “CDC” Centers for Disease Control and Prevention (CDC) miniature light traps.

Line 273: Spell out “BG” Biogents (BG) lures.

Line 276: Supposed to be “there are no known studies that have been”

Author Response

Response to Reviewer 3 Comments

Point 1: This is a very well written manuscript with a comprehensive experimental design and relevant analysis. The article does need to be combed through for standardizing acronym use throughout.

Response:  Thank you.

Point 2: Line 43-45, Line 49-51: This reads redundantly for me. I would revisit these two sentences and combine/revise.

Response: Thank you for the suggestion. Revised accordingly for the combine. 

Point 3: Line 59: It looks like this should be “, depending on the type of battery.”

Response: Thank you for the suggestion. It has been corrected.

Point 4: Line 63: Spell out UV on first use

Response: Corrected.

Point 5: Line 66: Are you using LT trap throughout the document? Either remove this or standardize throughout.

Response: Corrected.

Point 6: Line 73: Spell out LED on first use

Response: Corrected.

Point 7: Line 84: Signify urban mosquitoes

Response: Corrected. It was erroneously phrased as ‘urban mosquitoes’ instead of ‘urban area’.

Point 8: Figure 1: Is it supposed to say “trap sites” above the light types?

Response: Corrected.

Point 9: Line 101: Lowercase T in trap

Response: Corrected.

Point 10: Line 129: Lowercase O in one-way

Response: Corrected.

Point 11: Figure 2: This needs some reformatting. I am not sure if including the values are necessary in this graph (they are visualized by the y-axis and written in text) but if the authors with to include them then the values should not overlap each other or the error bars. Table 1: This table is cluttered and difficult to visualize. Add some space between mosquito species. “Trap light source” and “Collection nights” labels are too cramped.

Response: Figure 2 the numbers have been moved up to make the numbers clearer. The layout was edited of the table 1. In its current form, it is very clear.

Point 12: Table 2 & 3: This table needs revisiting. It is unclear what “*”, “**”, “***”, “a” and “b” means.

Response: Thank you for the suggestion. Table 2 & 3: for “*”, “**”, “***”, “a” and “b”. It has been corrected.

Point 13: Line 175: Italics Cx. quinquefasciatus

Response: Corrected.

Point 14: Line 176: Italics Cx. gelidus

Response: Corrected.

Point 15: Line 217-218: I suggest this sentence be removed; it appears to suggest that light traps can be used to reduce vector densities. If the author is suggesting that this is the case, then please provide a citation that this is possible.

Response: Corrected.

Point 16: Line 223: If you are beginning a sentence, spell out the entire genus.

Response: Thank you for the suggestion. It has been corrected.

Point 17: Line 270: Spell out “CDC” Centers for Disease Control and Prevention (CDC) miniature light traps.

Response: Corrected.

Point 18: Line 273: Spell out “BG” Biogents (BG) lures.

Response: Corrected.

Point 19: Line 276: Supposed to be “there are no known studies that have been”

Response: Thank you for the suggestion. It has been incorporated.

Response: Thank you. Revised accordingly.
